# Mediterranean Diet Adherence beyond Boundaries: Sociodemographic and Pregnancy-Related Determinants among Saudi Women

**DOI:** 10.3390/nu16152561

**Published:** 2024-08-04

**Authors:** Heba A. Ibrahim, Majed S. Alshahrani, Wafaa T. Elgzar

**Affiliations:** 1Department of Maternity and Childhood Nursing, Nursing College, Najran University, Najran 66441, Saudi Arabia; heaibrahim@nu.edu.sa; 2Department of Obstetrics and Gynecology, Faculty of Medicine, Najran University, Najran 66441, Saudi Arabia

**Keywords:** adherence, determinants, mediterranean diet, sociodemographic, pregnancy

## Abstract

Although the expected benefits of the Mediterranean diet (MD) are comprehensive, its implementation is hampered by poor adherence. Several factors can affect adherence to MD guidelines. The current study aimed to explore sociodemographic and pregnancy-related determinants of MD adherence among Saudi women. A correlational cross-sectional research design was conducted on a snowball sample of 774 pregnant women from the Najran region, Saudi Arabia, using an online survey between February and May 2024. A self-administered questionnaire consisting of sociodemographic data, pregnancy-related characteristics, and the MD scale was used for data collection. The current study showed that only 32.2% of participants had high adherence to the MD, and 57.6% had moderate adherence. Regarding sociodemographic determinants of MD adherence, highly educated, older women with lower pre-pregnancy body mass index (BMI) and higher monthly income increased the probability of high adherence to the MD (*p* < 0.05). In addition, being physically active before or during pregnancy significantly increased the woman’s probability of having higher adherence to the MD (*p* < 0.05). Concerning pregnancy-related determinants, having a planned pregnancy and regular antenatal care (ANC) increased the woman’s probability of high adherence to the MD by nearly 1.3 times (*p* < 0.05). In addition, low adherence to the MD increases the risk of gestational diabetes. In conclusion, numerous sociodemographic and pregnancy-related determinacies can significantly affect a woman’s adherence to the MD. Healthcare providers should address these determinants during the planning and implementation of pregnant women’s nutritional counseling to make the counseling process woman-centered and more effective.

## 1. Introduction

Gestation is a special period of a woman’s life that numerous health and lifestyle-related factors may influence. Preserving healthy dietary habits as good nutrition is crucial, particularly among pregnant women, as it supports the mother’s well-being and provides the necessary nutrition for the proper growth and development of the fetus. Nutritional insufficiencies can be aggravated during pregnancy due to added nutritional requirements supporting fetal growth and development [1]. The inability to meet these increased requirements is correlated with possibly harmful impacts on the pregnant woman and her fetus, such as intrauterine fetal growth restriction, cardiac disease, gestational diabetes, and anemia [2]. Maternal malnutrition can affect offspring’s health in the short- and long-term [3]. The Mediterranean diet (MD) is a healthy dietary style that can meet pregnant women’s requirements [4].

The Mediterranean diet has been verified to have a protective effect against obesity, metabolic and cardiovascular disorders, and malignancies due to a higher intake of antioxidants and a lower intake of saturated fats [5]. The MD focuses on the increased and sustainable consumption of unprocessed foods, such as vegetables, fruits, nuts, legumes, whole grains, fish, olive oil, and moderate amounts of dairy products, fish, meat, and poultry [6]. The MD is known as the healthiest diet style in the world [7]. In contrast, the intake of the Western diet is considered unhealthy; it includes red meat, fats, refined carbohydrates, sugars, and salt [8].

Evidence indicates that the MD is linked with numerous health benefits through all stages of life [9], including intrauterine life and fetus development. These benefits include a reduced risk of excessive weight gain and iron deficiency anemia during pregnancy [4,10], decreased incidence of gestational diabetes mellitus in moms and congenital anomalies in offspring [11,12], reduced risk of intrauterine growth retardation and neural tube defects [13]; and reduction of the risks of cardiac and metabolic diseases in offspring [14].

Although the expected benefits of the MD are comprehensive, its implementation is hampered by poor adherence [15]. Several factors can affect adherence to MD guidelines. Cultural, social, and economic factors, including urbanization, have caused people globally to move from traditional healthy diets to unhealthy Western diets [16]. In addition, lower educational levels and nutritional knowledge have also been correlated to poor adherence to the MD [15,17].

Dietary patterns were studied among Saudi individuals in different regions and age groups and for both genders [18,19]. The results revealed an increased consumption of highly processed food and sugar-sweetened drinks and a lower consumption of dairy products and whole grains, especially among young people. Data from the Saudi Health Survey indicated that only 5.2% of people met nutritional guideline recommendations for fruits, 7.5% for vegetables, 44.7% for fish and seafood, and 31.4% for nuts [20,21]. The prevalence of obesity among pregnant women in Saudi Arabia is increasing (68%) and is associated with pregnancy-related complications, such as gestational diabetes [22]. Furthermore, obesity affects the chance of pregnancy and may reduce responses to fertility treatment [22].

To our knowledge, research on adherence to the MD is mostly conducted in Mediterranean countries, and its adherence in non-Mediterranean countries is mostly unknown, especially among vulnerable groups, such as pregnant women. Limited research has evaluated MD adherence in Saudi Arabia [23]. However, no previous studies have explored adherence to the MD among pregnant women. Moreover, a literature gap exists on the sociodemographic and pregnancy-related factors associated with MD adherence in Saudi Arabia. Therefore, identifying these specific factors affecting maternal dietary practice and adherence to the MD is necessary and important to design proper interventions. These interventions can significantly enhance MD adherence even in different cultures and beyond Mediterranean boundaries. Hence, the current research was conducted to explore sociodemographic and pregnancy-related determinants of MD adherence in Saudi Arabia.

## 2. Subjects and Methods

Research Design: A correlational cross-sectional research design was utilized in the current study between February and May 2024.

Setting: The current study was conducted using an online questionnaire targeting pregnant women in the Najran region, Saudi Arabia. The pregnant women’s involvement was boosted through social networks such as Facebook, WhatsApp, Twitter, and Snapchat.

Subjects: Saudi pregnant women were recruited from the Najran community using the snowball sampling technique. First, pregnant women were accessed from the antenatal clinic at the Maternal and Children Hospital and asked to fill out the online questionnaire, then pass the survey link to their pregnant women relatives and friends and to pregnant women groups. The required sample size was determined using the flowing formula:$$N=\frac{Z_{\alpha /2}^{2}\times P\times (1-P)\times D}{E^{2}}$$
where Z_α/2_ is 1.96 for alpha 0.05, as the pregnant women’s adherence to the Mediterranean diet is unknown in Saudi Arabia; therefore, its proportion (P) will be considered (50%). D is the design effect (2), and E is the precision (or margin of error) (0.05). The proposed sample size was 768, and 5% was added to replace the case drop due to inconvenient data. After checking data quality, 32 cases were excluded due to inconvenient data, and the final statistical analysis was conducted on 774 participants. Participants in the current study had to be ≥18 years old, Saudi citizens, and able to read, write, and interact with social media.

### Tool of Data Collection

An online self-administered questionnaire, which consisted of three sections, was used for data collection. Section I: Sociodemographic data. It collected data such as age; residence; weight before pregnancy; height; educational status; basic education (grade 1 to grade 9), secondary education (grade 10 to grade 12), university education, or postgraduates; employment status; and family income (not enough < 5000 SR; enough 5000–10,000 SR.

Enough and save > 10,000 SR and daily physical activities: The woman is considered physically active if she practices at least 30 min of any sport, including walking daily. Section II: Pregnancy-related characteristics: It involves gravidity, parity, gestational age, planning for the current pregnancy, regularity of ANC, and complications during the current pregnancy. Section III: Mediterranean Diet Scale: The original version of the Mediterranean diet scale was developed by Martínez-González et al. in Spanish language [13] and then translated into English [24]. Aljehani et al. translated the scale into the Arabic version and confirmed good internal validity (0.74). The Arabic version of the scale is composed of 13 items categorized into 4 domains: meat and processed foods (5 items); olive oil and sauce (3 items); fruits, vegetables, nuts, and legumes (4 items); and fish and seafood (1 item). All items were dichotomous and scored 0 or 1 based on the adherence to MD. The total MD adherence was calculated by summing the scale items, then it was categorized as low (<5), moderate 5–10), and high (>10) adherence based on the total score [25].

Ethical Considerations: This study was approved by the deanship of scientific research and the ethical committee at Najran University (I.R.B. registration number 202403-076-018912-042978). The research purpose was elaborated on at the beginning of the online questionnaire, and informed consent was required before accessing the questionnaire. Women’s anonymity was applied, and data confidentiality was considered, as it was used only for research purposes.

Data analysis: IBM version 25 was utilized for data analysis. The data were cautiously evaluated for inconsistency, and 32 cases were excluded. Then, the data were tested for normal distribution. Descriptive statistics, such as frequencies, percentages, means, and standard deviations, were performed to represent participants’ demographic data, pregnancy-related characteristics, and adherence to MD. Logistic regression analyses were performed to examine the sociodemographic and pregnancy-related factors associated with high adherence to MD. Hosmer–Lemeshow and Omnibus tests were used to test the model goodness of fit. The multi-co-linearity test assessed the correlation between demographic and pregnancy-related variables via variance inflation factor and standard error; no variables were observed with variance inflation factor of >10 and standard error > 2. The direction and strength of statistical relationships were assessed by the adjusted odds ratio (AOR) with 95% confidence interval (CI). The significance level was considered at *p* < 0.05.

## 3. Results

Sociodemographic characteristics

The study results show that around three-quarters (71.1%) of the study participants were aged <35 years, with a mean age of 27.4 years. Being overweight before pregnancy was present among 61.5%, with a mean pre-pregnancy BMI of 26.35. In addition, 81.3% of the participants were university or postgraduate educated. Nearly a similar percentage of the study participants were urban area residents (77.4%), had insufficient monthly income (74.8%), and were housewives (76.6%). Around two-thirds of the participants were physically active before pregnancy (60.7%), and only 37.0% continued their physical activities after pregnancy (Table 1).

2.Pregnancy-related determinants

Regarding pregnancy-related determinants, 64.5% of the study participants were multi-gravida; the mean gravidity, parity, and gestational age were 1.8, 1.3, and 16.6, respectively. Moreover, 60.6% of the study participants planned for the current pregnancy, and 82.9% regularly attended antenatal visits. Nearly an equal proportion of the study participants suffered from pregnancy-induced hypertension (3.6%) and gestational diabetes (3.0%); in addition, 14.2% suffered from anemia (Table 2).

3.Participants’ adherence to MD

Adherence to the MD is discussed under four main categories. First, meat and processed foods were discussed: Around one-quarter of the study participants eat less than one tablespoon of butter, hydrogenated margarine, or cream (24.4%) and drink less than one serving of sweet or sweetened drinks (27.9%) daily. Nearly one-half of the study participants eat poultry more often than meat (45%) and limit red and processed meats to one serving or less one or two times weekly (50.1%). In addition, 60.1% eat less than three servings of sweets or pastries weekly. Second, olive oil and sauce were discussed: Around three-quarters use at least four tablespoons or more of olive oil daily (72.1%) and add some flavorings to food two or more times a week (82.6%). In addition, 39.9% use olive oil as the main fat source when cooking. Third, fruits, vegetables, nuts, and legumes were discussed: A large proportion of the study participants eat two servings or more of vegetables daily (75.1%) and three servings or more of fruit (62.3%). In addition, over one-third eat three servings or more of legumes (46.5%) and one serving or more of nuts (37.0%) weekly. Fourth, fish and seafood were discussed: Slightly more than one-half (54.1%) of the study participants eat three servings or more of fish or seafood each week. Generally, only 32.2% of the study participants had high adherence to the MD, and more than half (57.6%) had moderate adherence (Table 3).

4.Sociodemographic and Pregnancy-Related Determinants of MD adherence

The logistic regression analysis showed numerous sociodemographic and pregnancy-related determinants of MD adherence. First, sociodemographic determinants included age, pre-pregnancy BMI, education, monthly income, employment status, and physical activities before and during pregnancy. In detail, a one-year increase in the participant’s age increases the probability of high adherence to the MD by 1.5 times [AOR = 1.520, CI 95% = (1.250–1.810), *p* = 0.000)]. On the contrary, a one-point increase in pre-pregnancy BMI decreased the probability of high adherence to the MD by 0.5 times [AOR = 0.436 CI 95% (0.221–0.860), *p* = 0.017)]. In addition, being university/postgraduate educated increased the probability of high adherence to the MD by 1.7 times when taking basic education as a reference [AOR = 1.720, CI 95% (1.143–2.588), *p* = 0.009)]. Having enough [AOR = 1.370, CI 95% (0.962–1.672), *p* = 0.029)] or more than enough [AOR = 1.350, CI 95% (0.943–1.770), *p* = 0.025)] monthly income and being a working woman [AOR = 1.290, CI 95% (1.040–1.550), *p* = 0.023)] increased the chance of having higher adherence to the MD by 1.3 times. In addition, being physically active before [AOR = 3.815, CI 95% (2.595–5.609), *p* = 0.000)] or during pregnancy [AOR = 4.339, CI 95% (2.875–6.547), *p* = 0.000)] significantly increased the woman’s probability to have higher adherence to the MD by 3.8 and 4.3 times, respectively. Second, pregnancy-related determinants include planning for the current pregnancy, ANC regularity, and complications during the current pregnancy. Specifically, having a planned pregnancy [AOR = 1.438, CI 95% (1.117–1.852), *p* = 0.005)] and regular ANC [AOR = 1.350, CI 95% (1.192–1.530), *p* = 0.000)] increased the woman’s probability to high adherence to the MD by nearly 1.35 times. In addition, low adherence to the MD increases the risk of gestational diabetes [AOR = 0.448, CI 95% (0.219–0.859), *p* = 0.018). (Table 4).

## 4. Discussion

The current study evaluates MD adherence among pregnant women in Saudi Arabia and explores its sociodemographic and pregnancy-related determinants. Overall, around one-third (32.2%) of the participants had high adherence to the MD, and more than half (57.6%) had moderate adherence. These findings may be attributed to a lack of awareness about the MD components and health benefits among pregnant women in Saudi Arabia. Our findings for MD adherence were comparable to previous studies in Mediterranean countries and non-Mediterranean countries, including Saudi Arabia. In Saudi Arabia, there are no studies that discuss MD adherence among pregnant women; however, a recent cross-sectional survey concluded that only one-fifth of the adult population adhered to the MD [26]. In the United Arab Emirates, a recent study showed that 41% of adult populations had moderate adherence to the MD, and only 23% had high adherence [27]. The MD adherence level in the two abovementioned studies is lower than the present one. The slight increase in MD adherence in the current study can be explained by the fact that pregnant women are usually more motivated to modify their dietary habits to respond to the pregnancy requirements and to ensure the health and well-being of themselves and their offspring. Consistent with this interpretation, Gardner et al. stated that the perceived benefits of adherence to a healthy diet enhanced pregnant women’s intentions to consume a large number of healthy foods, such as fruits and vegetables, and decreased their consumption of high-fat foods during gestation [28].

Moderate adherence to the MD has also been reported from adult Arab females of fourteen Arab nationalities residing in Jordan [29]. In addition, two Spanish studies showed that high adherence to the MD among university students ranged from 24 to 36% [30,31]. In the same line, another Spanish cohort study recruited 738 pregnant participants and reported that 28.5% of the pregnant women were highly adherent to the MD [32]. Moreover, only one-quarter of the sample of pregnant women from ten distinct Greek regions had high adherence to the MD [33]. These results highlighted the necessity to carry out effective educational interventions that support healthy eating habits and increase pregnant women’s awareness regarding the MD.

Regarding sociodemographic determinants, the current study showed that higher maternal adherence to the MD during pregnancy was associated with age. This means that increasing participants’ age by one year increases the probability of high adherence to the MD by 1.5 times. The scientific literature revealed a positive association between older maternal age and greater adherence to the MD and healthy dietary patterns [33,34]. The association between age and adherence to the MD pattern may be attributed to older pregnant women being more likely to have a planned pregnancy and better nutritional knowledge. Therefore, they were more likely to eat healthier food and adhere better to the MD to prepare for pregnancy, hence a higher diet quality [14].

In addition, the current study demonstrated that educational level was among the sociodemographic factors that affect the probability of obtaining a high score for MD adherence. Being university or postgraduate educated increased the probability of high adherence to the MD by 1.7 times when taking basic education as a reference. In accordance with our findings [33,35], we tried to explore the determinant of high adherence to the MD in Greece and Italy. They confirmed the role of education in adherence to the MD, as a highly educated participant showed higher MD adherence and a higher consumption of healthy foods. Another study on pregnant women in Central South Africa indicated that greater compliance with the MD was associated with an advanced educational level [36]. Therefore, we can conclude from this result that higher levels of education predict better nutritional literacy in terms of a healthy diet and food quality. Therefore, one important strategy to improve pregnant women’s nutrition is to give more attention to girls’ education, who are the future mothers.

Other determinants of MD adherence were a higher monthly income and occupational status. Having enough or more than enough monthly income and being a working woman increased the chance of having higher adherence to the MD by 1.35 times. These results are in agreement with other similar studies. A study conducted in Greece on 5688 pregnant women by Antasouras et al. found that higher adherence to the MD was associated with higher economic status [33]. In addition, an Italian study investigated MD adherence among 838 adult populations and found that having high economic status is a very important factor that enhances not only high but sustainable adherence to the MD [35]. The high cost of a healthy diet mostly explains the link between high economic status and high sustainable adherence to the MD [37]. In some instances, people may have knowledge and motivation for a healthy diet, but limited financial resources restrict their adherence.

In the present study, BMI was negatively associated with MD adherence. In other words, a one-point increase in pre-pregnancy BMI decreased the probability of high adherence to the MD by 0.5 times. This result is consistent with previous studies that suggested that pre-pregnancy BMI was inversely related to higher MD adherence and diet quality [38]. It is essential to highlight that a poor pre-pregnancy diet quality may contribute to increased pre-pregnancy BMI. As a result, it may be possible that overweight and obese pregnant women have poorer diet quality and, thus, lower MD adherence before pregnancy, which also continues during pregnancy. In this context, it has been reported by Olmedo-Requena et al. that dietary habits in eating foods such as fish, fruits, and vegetables remained similar during gestation compared to the period before gestation [39].

Another important result was the significant relation between physical activity and the MD adherence score. Our findings revealed that being physically active before or during pregnancy significantly increased the woman’s probability of having higher adherence to the MD by 3.8 and 4.3 times, respectively. These findings were similar to the findings of other researchers [32,38], who concluded that pregnant women who engage in physical activity during their free time are more committed to the MD and make healthy food choices compared to those who do not engage in physical activity. Savard et al. reported that the best determinant of poor diet quality during gestation was low physical activity levels [40]. The positive relationship between physical activity and MD compliance is noteworthy, suggesting that educational interventions should address appropriate healthy diet and physical activity in conjunction [41].

Concerning the pregnancy-related determinants, our findings provide new insights into pregnancy-related determinants of high MD adherence. The present findings indicated that having a planned pregnancy and regular ANC increased the woman’s probability of high adherence to the MD by nearly 1.3 times. It is important to highlight the ANC’s role in improving knowledge and promoting positive attitudes towards healthy nutrition and diet quality, which affect the intention to change their diet, especially for younger and low-educated women. The association between the regularity of ANC, pregnancy planning, and MD adherence is rarely investigated. However, the findings from a German study that included 123 pregnant women by Ehrhardt et al. [42] reported that pregnant women consider nutritional counseling during prenatal follow-up to be crucial to improving diet quality. They further recommended intensifying nutritional advice during pregnancy in order to promote healthy diet behaviors. Therefore, implementing effective prenatal classes and individual nutritional counseling in primary health centers is necessary to increase adherence to a healthy dietary pattern, such as the MD, during pregnancy. Focus should be placed on increasing the consumption of beneficial foods, such as fruits, vegetables, pure olive oil, fish, dairy products, nuts, etc., in addition to reducing the consumption of sugary drinks, processed meats, pastries, hydrogenated oils, etc. Using this dietary pattern in a preventive manner can be a beneficial and low-cost health strategy to avoid overweight body types and pregnancy complications such as gestational diabetes and nutrient deficiencies.

Our study findings indicated that low adherence to the MD increases the risk of gestational diabetes. In accordance with the current study finding, a recent observational study on 193 low-risk pregnant women in Greece stated that greater adherence to the MD throughout gestation was associated with lower gestational diabetes risk [43]. In addition, a case-control study conducted by Olmedo-Requena et al. indicated that greater compliance with MD pre-pregnancy was associated with gestational diabetes reduction [41]. These aforementioned results strengthen previously identified relations and highlight the importance of changing unhealthy eating habits and their potential impact during pregnancy in controlling and reducing the risk of pregnancy metabolic complications, such as gestational diabetes. Given the importance of a healthy diet during pregnancy, effective interventions in primary care centers to promote MD adherence are necessary. Attention should also be paid to ongoing individual nutritional counseling in order to enhance nutrition; increase intakes of dietary fiber, fruits, and vegetables; and reduce intakes of saturated fats.

### Strengths and Limitations

The current study is the first one to address adherence to the MD among pregnant women in Saudi Arabia. Adherence to the MD was assessed using a standardized tool that has been validated in Saudi Arabia; however, the tool does not take into account important food groups such as dairy products and cereals. The self-reported nature of the questionnaire increased the risk of memorization bias or reporting data for social acceptance.

## 5. Conclusions

The current study showed that around one-third of the study participants had high adherence to the MD. Regarding sociodemographic determinants of MD adherence, highly educated, older women with lower pre-pregnancy BMI and higher monthly income had higher odds of MD adherence. In addition, being physically active before or during pregnancy significantly increased the woman’s probability of having higher adherence to the MD. Concerning pregnancy-related determinants, having planned pregnancy and regular ANC increased the woman’s probability of high adherence to the MD. In addition, women who have gestational diabetes have a lower tendency to adhere to the MD. Healthcare providers should address these determinants during the planning and implementation of pregnant women’s nutritional counseling to make the counseling process woman-centered and more effective in increasing adherence to the MD.

## Figures and Tables

**Table 1 nutrients-16-02561-t001:** The sociodemographic characteristics of the participants. (*n* = 774).

Sociodemographic Data	N	%
**Age in years**		
**<35**	550	71.1
**≥35**	224	28.9
**Mean ±SD**	27.41 ± 5.68 years	
**Pre-pregnancy BMI, kg/m^2^**		
Underweight (BMI ˂ 18.5)	38	4.9
Normal weight (BMI = 18.5 ˂ 25)	145	18.7
Overweight (BMI =25 ˂ 30)	476	61.5
Obese (BMI = 30 and more)	115	14.9
**Mean ±** **SD**	26.35 ± 10.37	
**Residence**		
Urban	599	77.4
Rural	175	22.6
**Education**		
Basic education (grade 1 to grade 9)	22	2.8
Secondary education (grade 10 to grade 12)	123	15.9
University education/postgraduate	629	81.3
**Monthly income**		
Not enough < 5000 SR	579	74.8
Enough 5000–10,000 SR	125	16.1
Enough and save >10,000 SR	70	9.0
**Employment status**		
Working	181	23.4
Housewife	593	76.6
**Physical activity before pregnancy**		
Yes	470	60.7
No	304	39.3
**Physical activity during the current pregnancy**		
Yes	286	37.0
No	488	63.0

**Table 2 nutrients-16-02561-t002:** Pregnancy-related determinants of the participants. (*n*= 774).

Reproductive Characteristics	N	%
**Gravidity**		
**Primi-gravida**	275	35.5
**Multi-gravida**	499	64.5
**Gravidity (Mean ±** **SD)**	1.81 ± 1.45	
**Parity (Mean ±SD)**	1.36 ± 0.73	
**Gestational age in weeks (Mean ±SD)**	16.60 ± 8.38 weeks	
**Planning to current pregnancy**		
Yes	469	60.6
No	305	39.4
**ANC Regularity**		
Yes	642	82.9
No	132	17.1
**Complications during the current pregnancy**		
Pregnancy-induced hypertension	28	3.6
Gestational diabetes	23	3.0
Anemia	110	14.2
Free from a pregnancy complication	613	79.2

**Table 3 nutrients-16-02561-t003:** Participants’ adherence to MD (*n* = 774).

Mediterranean Diet Subscale	Items	N	%
**Meat and processed foods**	Eat less than one tablespoon of butter, hydrogenated margarine, or cream each day	189	24.4
	Drink less than one serving of sweet or sweetened drinks each day	216	27.9
	Eat poultry (chicken or turkey) more often than meat (beef, veal, hamburger, or sausage)	348	45.0
	Limit red meat and processed meats to one serving or less one or two times a week	388	50.1
	Eat less than three servings of sweets or pastries each week	465	60.1
**Olive oil and Sauce;**	Use olive oil as the main source of fat when you cook	309	39.9
	Use at least four tablespoons or more of olive oil when cooking your food each day	558	72.1
	Add some flavorings to your food, such as a mixture of tomatoes, garlic, onions, and leeks, two or more times a week	639	82.6
**Fruits, vegetables, nuts, and legumes**	Eat two servings or more of vegetables each day	581	75.1
	Eat three servings or more of fruit each day	482	62.3
	Eat three servings or more of legumes a week	360	46.5
	Eat one serving or more of nuts each week	286	37.0
**Fish and seafood**	Eat three servings or more of fish or seafood each week	419	54.1
**Total score**	Low adherence	79	10.2
	Moderate adherence	446	57.6
	High adherence	249	32.2

**Table 4 nutrients-16-02561-t004:** The ordinal logistic regression analysis for high adherence to the MD.

Variable	COR (95% CI)	*p*-Value	AOR (95% CI)	*p*-Value
**Age in years**	1.499 (1.221–1.781)	0.000 *	1.520 (1.250–1.810)	0.000 *
**Pre-pregnancy BMI, kg/m^2^**	0.335 (0.219–0.873)	0.015 *	0.436 (0.221–0.860)	0.017 *
**Residence**				
Urban	1.624 (0.841–3.134)	0.148	1.498 (0.727–3.086)	0.273
Rural	Ref		Ref	
**Education**		0.020 *		0.031 *
Basic education (grade 1 to grade 9)	Ref		Ref	
Secondary education (grade 10 to grade 12)	0.906 (0.327–2.513)	0.850	0.877 (0.318–2.415)	0.799
University education	1.783 (1.179–2.694)	0.006 *	1.720 (1.143–2.588)	0.009 *
**Monthly income**		0.002 *		0.026 *
Not enough < 5000 SR	Ref		Ref	
Enough 5000–10,000 SR	2.021 (1.306–3.126)	0.002 *	1.370 (0.962–1.672)	0.029 *
Enough and save > 10,000 SR	1.925 (1.253–2.957)	0.003 *	1.350 (0.943–1.770)	0.025 *
**Employment status**				
Housewife	Ref		Ref	
Working	2.003 (1.297–3.095)	0.002 *	1.290 (1.040–1.550)	0.023 *
**physical activity before pregnancy**				
Yes	3.638 (2.468–5.360)	0.000 *	3.815 (2.595–5.609)	0.000 *
No	Ref		Ref	
**physical activity during the current pregnancy**				
Yes	4.870 (2.569–9.232)	0.000 *	4.339 (2.875–6.547)	0.000 *
No	Ref		Ref	
**Gravidity**	1.126 (1.013–1.251)	0.027 *	0.989 (0.787–1.244)	0.925
**Parity (Mean ±SD)**	1.284 (1.129–1.461)	0.000 *	1.552 (.828–2.907)	0.170
**Gestational age in weeks (Mean ±SD)**	0.865 (0.612–1.137)	0.135	0.717 (0.491–1.046)	0.084
**Planning to current pregnancy**				
Yes	1.948 (1.273–2.983)	0.002 *	1.438 (1.117–1.852)	0.005 *
No	Ref		Ref	
**ANC Regularity**				
Yes	1.811 (1.058–2.672)	0.000 *	1.350 (1.192–1.530)	0.000 *
No	Ref		Ref	
**Complications during the current pregnancy**		0.0228 *		0.0425 *
Pregnancy-induced hypertension	0.752 (0.225–1.615)	0.423	0.941 (0.232–1.711)	0.674
Gestational diabetes	0.308 (0.146 –0.650)	0.002 *	0.448 (0.219–0.859)	0.018 *
Anemia	0.432 (0.215–0.869)	0.019 *	0.709 (0.501–1.052)	0.091
Free from a pregnancy complication	Ref		Ref	

A.O.R.: Adjusted odds ratio. CI: Confidence interval * significant at *p* ˂ 0.05.

## Data Availability

The corresponding author will make data available upon reasonable request.

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
