# Peer review of "Mediterranean Diet Adherence beyond Boundaries: Sociodemographic and Pregnancy-Related Determinants among Saudi Women"

_nutrients, 2024, doi:10.3390/nu16152561_

Round 1

Reviewer 1 Report

Comments and Suggestions for Authors

TO THE AUTHORS

Brief Summary

This cross-sectional study investigated sociodemographic and perinatal factors associated with high adherence to the Mediterranean diet (MD) among women residing in Saudi Arabia. Results showed that overall, Saudi Arabian women have low adherence to MD and that being highly educated, and older, having lower pre-pregnancy BMI and higher monthly income were factors that significantly increased the odds of high adherence to MD. The authors concluded that healthcare providers should consider these factors when implementing interventions promoting MD adherence during pregnancy.

Indeed, whether the benefits of the Mediterranean diet can be translated to a non-Mediterranean population is worth consideration and would be of interest to health professionals. Overall, a good effort was undertaken by the authors. However, there are methodological errors that require revision before this study is scientifically sound. Regarding the methods section, more details about the sociodemographic questionnaire are required. Study limitations and strengths were not mentioned, and this is a downfall of this manuscript. Please see this as an opportunity to master the art of scientific writing. We look forward to more studies from this research group.

Please refer to my comments below.

COMMENTS

ABSTRACT

-Line 12 Sociodemographic, typo error small ‘s’

-MAIN MANUSCRIPT

Line 46-48. ‘MD focuses on increased and sustainable consumption of unprocessed foods such as vegetables, fruits, nuts, legumes, whole grains, fish, olive oil, and moderate amounts of dairy products, fish, meat, and poultry’

Add reference (ref) at the end of this statement

-Line 48-49 Add ref ‘MD is known as the healthiest diet style in the world’

-Line 64 ‘Dietary patterns were studied among Saudi individuals in different regions and age groups and for both genders’

Add ref to the study that you are referring to.

-Line 69-71,  ‘The prevalence of obesity among pregnant women in Saudi Arabia is increasing (68%) and is associated with pregnancy-related complications such as gestational diabetes’

Add ref

METHODS

-Lines 87- 88 ‘The current study was conducted using an online questionnaire targeting pregnant women in the Najran region, Saudi Arabia.’

Add an internet link where the online questionnaire can be found, or add the questionnaire in the online supplement. A reference to the protocol study which describes the study in detail [objectives, population and materials used, etc.], is also needed.

Assessment Tools

Describe the sociodemographic factors that were assessed,  in terms of the categories for each variable. For example, in Saudi Arabia, how do you define basic education, secondary, etc.?

-Instead of ‘working conditions’, employment status would be more appropriate

-Line 152 for gestational age 16.6, add the units i.e., ‘years’

-In the headings of tables 1-3, if ‘No’ is abbreviated for number, it is preferable to refer to numbers/or counts as ‘n’ and ‘N’ for the total sample of participants. Please revise.

Logistic regression model

-In both the text and below Table 4, mention the confounding factors that were used in the adjusted regression analysis

-Statistically significant p-values can be shown in bold.

-In Table 4, add the ORs and p-values from the crude (unadjusted) analysis in a column before the adjusted.

DISCUSSION

-Lines 261-263, The high cost of a healthy diet mostly explains the link between high economic status and high sustainable adherence to MD

Add a reference to support that a healthy diet is more expensive.

STRENGTHS/LIMITATIONS

- In terms of study transparency, it is important that the study's strengths and limitations are mentioned before the conclusions.

Author Response

For research article

Response to Reviewer (1) Comments

1. Summary

Thank you very much for taking the time to review this manuscript. Please find the detailed responses below and the corresponding revisions/corrections highlighted in the re-submitted files.

2. Questions for General Evaluation

Reviewer’s Evaluation

Response and Revisions

Does the introduction provide sufficient background and include all relevant references?

Yes/Can be improved/Must be improved/Not applicable

Are all the cited references relevant to the research?

Yes/Can be improved/Must be improved/Not applicable

Is the research design appropriate?

Yes/Can be improved/Must be improved/Not applicable

Are the methods adequately described?

Yes/Can be improved/Must be improved/Not applicable

The modification has been done

Are the results clearly presented?

Yes/Can be improved/Must be improved/Not applicable

Are the conclusions supported by the results?

Yes/Can be improved/Must be improved/Not applicable

3. Point-by-point response to Comments and Suggestions for Authors

Comments 1: [ABSTRACT -Line 12 Sociodemographic, typo error small ‘s’.]

Response 1: [corrected] Thank you for pointing this out. We agree with this comment. Therefore, we corrected the text in the manuscript to “sociodemographic. This change can be found – page number1, paragraph (under the abstract), and line 12.

Comments 2: [-Line 48-49 Add ref ‘MD is known as the healthiest diet style in the world’.]

Response 2: Agree. We added references that emphasize this point. – page number 2, 1st paragraph, and line 49.

Comments 3: [--Line 64 ‘Dietary patterns were studied among Saudi individuals in different regions and age groups and for both genders’. Add ref to the study that you are referring to.]

Response 3: Agree. We have added references that emphasize this point. – page number 2, 3rd paragraph, and line 65.

Comments 4: [--Line 69-71, ‘The prevalence of obesity among pregnant women in Saudi Arabia is increasing (68%) and is associated with pregnancy-related complications such as gestational diabetes’ Add ref.]

Response 4: Agree. We added references that emphasize this point. – page number 2, 3rdparagraph, and line 72.

Comments 5: [--Lines 87- 88 ‘The current study was conducted using an online questionnaire targeting pregnant women in the Najran region, Saudi Arabia.’

Add an internet link where the online questionnaire can be found, or add the questionnaire in the online supplement. A reference to the protocol study which describes the study in detail [objectives, population and materials used, etc.], is also needed.]

Response 5: Agree. We, added the questionnaire as a supplementary material to be published. The description of the study in detail (objectives, population and materials used) is present in the methodology section in details  

Comments 6: [-Assessment Tools: Describe the sociodemographic factors that were assessed, in terms of the categories for each variable. For example, in Saudi Arabia, how do you define basic education, secondary, etc.?.]

Response 6: Agree. We defined the education and family income categories in Saudi Arabia in the assessment tools (methodology section) page number 3, 2nd paragraph, and line 110-112 and table 1 and 4.

Comments 7: [-Instead of ‘working conditions’, employment status would be more appropriate.]

Response 7: Agree. We replaced working conditions with employment status throughout the document.

Comments 8: [-Line 152 for gestational age 16.6, add the units i.e., ‘years’.]

Response 8: Agree. The unit (weeks) was added to clarify this point. Page number 5, table 2.

Comments 9: [-In the headings of tables 1-3, if ‘No’ is abbreviated for number, it is preferable to refer to numbers/or counts as ‘n’ and ‘N’ for the total sample of participants. Please revise.]

Response 9: Agree. We modified the abbreviation “No” to “N” in the headings of tables 1-3.

Comments 10: [-Logistic regression model -In both the text and below Table 4, mention the confounding factors that were used in the adjusted regression analysis.]

Response 10: Agree. We  added to the statistical analysis “The multi-co-linearity test assessed the correlation between demographic and pregnancy-related variables via variance inflation factor and standard error; no variables were observed with variance inflation factor of > 10 and standard error > 2” to emphasize this point. This change can be found – page number 3, 5th paragraph, and line 137-139.

Comments 11: [-Statistically significant p-values can be shown in bold.]

Response 11: Agree. We put all the significant values ​​in bold. Page number 7,8, Table 4.

Comments 12: [-In Table 4, add the ORs and p-values from the crude (unadjusted) analysis in a column before the adjusted.]

Response 12: Agree. We have, accordingly, added the crude (unadjusted) analysis in a column before the adjusted to emphasize this point. The changes made can be found in table 4– page 7,8

Comments 13: [--Lines 261-263, The high cost of a healthy diet mostly explains the link between high economic status and high sustainable adherence to MD. Add a reference to support that a healthy diet is more expensive.]

Response 13: Agree. We added a reference that support that a healthy diet is more expensive. – page number 9, 2nd paragraph, and line 266.

Comments 14: [-STRENGTHS/LIMITATIONS- In terms of study transparency, it is important that the study's strengths and limitations are mentioned before the conclusions.]

Response 14: Agree. We have, accordingly, added strength and limitation to emphasize this point. The study strength and limitation can be found – page number10, paragraph2, and line323-327.]

4. Response to Comments on the Quality of English Language

Point 1: English language fine. No issues detected

Response 1:    (in red)

5. Additional clarifications

[Thanks for your efforts .]

Reviewer 2 Report

Comments and Suggestions for Authors

OVERALL EVALUATION

The manuscript is interesting, but I have doubts regarding the choice of the questionnaire for determining adherence to the Mediterranean Diet.

 The questionnaire proposed by Ghisi et al (Reference n° 21) does not consider the consumption of dairy products (rarely present, but not absent in MD) and cereals, which instead, in the form of bread, pasta, pizza, biscuits, are present in the Mediterranean Diet (see Bach-Faig et al., 2011). This questionnaire does not even consider the consumption of wine, but since the study was conducted in an Islamic Country, the consumption of alcoholic beverages is probably zero.  A good questionnaire for evaluating adherence to the Mediterranean diet was proposed by Sofi et al (2017) which considers all the DM factors listed by Bach-Faig et al. (2011).  In my opinion, the fact of having chosen a Med Score which has many limitations, because it does not consider important food groups, should be discussed in the discussion.  In any case, the work can be accepted with minor revision.

 References

Sofi et al (2017) Validation of a literature-based adherence score to Mediterranean Diet: the MEDI-LITE score. Int J Food Sci Nutr. 2017 Sep;68(6):757-762. doi: 10.1080/09637486.2017.1287884

Bach-Faig A et al (2011) Mediterranean Diet Foundation Expert Group. Mediterranean diet pyramid today. Science and cultural updates. Public Health Nutr. 2011 Dec;14(12A):2274-84. doi: 10.1017/S1368980011002515.

MINOR REMARKS

Line 198-199. In my opinion, the sentence should be reversed. It is having a low adherence to the Mediterranean Diet that increases the risk of gestational diabetes.

Author Response

For research article

Response to Reviewer (2) Comments

1. Summary

Thank you very much for taking the time to review this manuscript. Please find the detailed responses below and the corresponding revisions/corrections highlighted in the re-submitted files.

2. Questions for General Evaluation

Reviewer’s Evaluation

Response and Revisions

Does the introduction provide sufficient background and include all relevant references?

Yes/Can be improved/Must be improved/Not applicable

Are all the cited references relevant to the research?

Yes/Can be improved/Must be improved/Not applicable

Is the research design appropriate?

Yes/Can be improved/Must be improved/Not applicable

Are the methods adequately described?

Yes/Can be improved/Must be improved/Not applicable

Are the results clearly presented?

Yes/Can be improved/Must be improved/Not applicable

Are the conclusions supported by the results?

Yes/Can be improved/Must be improved/Not applicable

3. Point-by-point response to Comments and Suggestions for Authors

Comments 1: [The manuscript is interesting, but I have doubts regarding the choice of the questionnaire for determining adherence to the Mediterranean Diet.

The questionnaire proposed by Ghisi et al (Reference n° 21) does not consider the consumption of dairy products (rarely present, but not absent in MD) and cereals, which instead, in the form of bread, pasta, pizza, biscuits, are present in the Mediterranean Diet (see Bach-Faig et al., 2011). This questionnaire does not even consider the consumption of wine, but since the study was conducted in an Islamic Country, the consumption of alcoholic beverages is probably zero. A good questionnaire for evaluating adherence to the Mediterranean diet was proposed by Sofi et al (2017) which considers all the DM factors listed by Bach-Faig et al. (2011).  In my opinion, the fact of having chosen a Med Score which has many limitations, because it does not consider important food groups, should be discussed in the discussion. In any case, the work can be accepted with minor revision.

Response 2: Agree. We appreciate your valuable comment, accordingly, we mentioned this point in the study limitation. Page number 10, paragraph, and line 324-328. However, the reason behind using Aljehani et al scale is that it was translated and validated in Saudi Arabia (the same setting of the current study)          

Comments 1: [MINOR REMARKS

Line 198-199. In my opinion, the sentence should be reversed. It is having a low adherence to the Mediterranean Diet that increases the risk of gestational diabetes.]

Response 2: Agree. We have, accordingly, reversed the statement to emphasize this point. Discuss Page number 6, paragraph 1, and line 200.        

4. Response to Comments on the Quality of English Language

Point 1: I am not qualified to assess the quality of English in this paper

Response 1:    (in red)

5. Additional clarifications

[Thanks for your efforts]
